# Preventable risk factors for type 2 diabetes can be detected using noninvasive spontaneous electroretinogram signals

Ramsés Noguez Imm[1]☉, Julio Muñoz-Benitez[2]☉, Diego Medina[2], Everardo Barcenas[2], Guillermo Molero-Castillo[2], Pamela Reyes-Ortega[1], Jorge Armando Hughes-Cano[1], Leticia Medrano-Gracia[3], Manuel Miranda-Anaya[4], Gerardo Rojas-Piloni[1], Hugo Quiroz-Mercado[5], Luis Fernando Hernández-Zimbrón[5,6], Elisa Denisse Fajardo-Cruz[5], Ezequiel Ferreyra-Severo[5], Renata García-Franco[7], Juan Fernando Rubio Mijangos[8], Ellery López-Star[8], Marlon García-Roa[8], Van Charles Lansingh[8], Stéphanie C. Thébault[1]*

1 Instituto de Neurobiología y Universidad Nacional Autónoma de México (UNAM), Campus UNAM-Juriquilla, Querétaro, Mexico, 2 Facultad de Ingeniería, Universidad Nacional Autónoma de México (UNAM), Ciudad Universitaria, Ciudad de México, Mexico, 3 IIMAS, Nacional Autónoma de México (UNAM), CDMX, Mexico, 4 Unidad Multidisciplinaria de Docencia e Investigación-Facultad de Ciencias, Universidad Nacional Autónoma de México (UNAM), Campus UNAM-Juriquilla, Querétaro, Mexico, 5 Research Department, Asociación Para Evitar la Ceguera, Mexico City, Mexico, 6 Clínica de Salud Visual, Escuela Nacional de Estudios Superiores, Unidad León, Universidad Nacional Autonóma de México (UNAM), León, Guanajuato, Mexico, 7 Instituto de la Retina del Bajío (INDEREB), Prolongación Constituyentes 302 (Consultorios 410 y 411, torre 3, Hospital San José), El jacal, Santiago de Querétaro, Querétaro, Mexico, 8 Instituto Mexicano de Oftalmología (IMO), I.A.P., Circuito Exterior Estadio Corregidora Sn, Centro Sur, Santiago de Querétaro, Querétaro, Mexico

☉ These authors contributed equally to this work.
* sthebault@comunidad.unam.mx

**Data Availability Statement:** All relevant data are within the paper and its Supporting Information files.

## Abstract

Given the ever-increasing prevalence of type 2 diabetes and obesity, the pressure on global healthcare is expected to be colossal, especially in terms of blindness. Electroretinogram (ERG) has long been perceived as a first-use technique for diagnosing eye diseases, and some studies suggested its use for preventable risk factors of type 2 diabetes and thereby diabetic retinopathy (DR). Here, we show that in a non-evoked mode, ERG signals contain spontaneous oscillations that predict disease cases in rodent models of obesity and in people with overweight, obesity, and metabolic syndrome but not yet diabetes, using one single random forest-based model. Classification performance was both internally and externally validated, and correlation analysis showed that the spontaneous oscillations of the non-evoked ERG are altered before oscillatory potentials, which are the current gold-standard for early DR. Principal component and discriminant analysis suggested that the slow frequency (0.4–0.7 Hz) components are the main discriminators for our predictive model. In addition, we established that the optimal conditions to record these informative signals, are 5-minute duration recordings under daylight conditions, using any ERG sensors, including ones working with portative, non-mydriatic devices. Our study provides an early warning system with promising applications for prevention, monitoring and even the development of new therapies against type 2 diabetes.

**Funding:** R.N.I. is a Doctoral student from the Programa de Posgrado en Ciencias, Universidad Nacional Autónoma de México (UNAM) and received fellowships from the National Council of Science and Technology of Mexico (CONACYT; #620199) and from UNAM DGAPA-PAPIIT #070122 and #189522. This study was supported by the UNAM grant IN209317 (ST), IN205420 (ST), CONACYT 299625 (ST), CONACYT CF-2019-1759 (ST), and Shedid grant (R. Miledí and A. Martínez Torres, acknowledgement since there are not co-authors). The funders had no role in study design, data collection and analysis, decision to publish, or preparation of the manuscript.

**Competing interests:** The authors have declared that no competing interests exist.

## Introduction

From a public health standpoint, timely detection of diabetic eye disease is one of the most cost-effective health procedures available to avoid the burden of vision loss [1]. Huge efforts are being made to screen for sight-threatening diabetic retinopathy (DR) [1, 2], but the best intervention remains to prevent the onset of diabetes [1].

DR affects nearly 100 million people worldwide [3], corresponding to approximately one-third of all people with diabetes. Type 2 diabetes accounts for 85%–95% of diabetes cases and it can be avoided often by healthy life choices [4]. Strategies to promote adherence to such choices would benefit from reliable, large-scale screening methods that identify and allow follow-up of people at risk of type 2 diabetes, i.e., those with overweight, obesity or metabolic syndrome, but still without diabetes. Ultimately, predicting risk factors of developing type 2 diabetes should mitigate the risk of developing complications like DR.

The screening method to be developed must overcome the limitations of the current gold standard tests, particularly the $A_1C$ [5] and the 2-hour post-challenge glucose tests [6], which are both invasive and require fasting. $A_1C$, in addition, does not always reflect or predict the burden of diabetes and fails to account for hypoglycaemia and glucose variability [7, 8]. The new screening method should also be based on a reliable biomarker of the silent installation of type 2 diabetes.

An appealing candidate is spontaneous retinal oscillations. Spontaneous neural oscillations have proven being biomarkers for neurodegenerative diseases [9] and diabetes is one of them [10]. Spontaneous brain activity changes with diabetes [11], but also with obesity [12–16]. Notably, retinal neurons can produce spontaneous activities [17–29] and neurodegenerative retinas showed altered patterns of spontaneous activity [30–32], including in diabetes conditions [33].

Here we propose a diagnostic prediction method for early risk factors of type 2 diabetes and thereby DR, based on the non-invasive recording of spontaneous retinal oscillations using a simple, yet meaningful non-evoked electroretinogram (ERG) protocol. ERG is the only clinical objective test recommended by the International Society for Clinical Electrophysiology of Vision to stage early DR [34], before vascular changes are apparent, and its clinical application has significantly improved with the commercial introduction of non-invasive, portative, and non-mydriatic ERG devices [35]. Nonetheless, ERG has always been based on the response of retinal cells to a flash of light and never under spontaneous conditions. We found that spontaneous ERG signals are differentially altered in rodent models of obesity and prediabetes, allowing their discrimination by a random forest-based prediction model. The model also predicts the evolution of the diabetes model and risk factors for DR in humans, including overweight, obesity, and metabolic syndrome. Our algorithm can be coupled with spontaneous ERG signals from different sensors. Principal component and discriminant analysis revealed slow ERG frequencies as main discriminators for our predictive model. Together, our study shows that spontaneous ERG signals are intimately linked to systemic metabolic status and demonstrates their use to screen people for preventable stages of DR.

## Materials and methods

### Ethics statement

All animal experiments were approved by the Bioethics Committee of the Institute of Neurobiology (protocol #74) at UNAM (clave NOM-062-ZOO-1999), which has jurisdiction to approve animal studies, in accordance with the rules and regulations of the Society for Neuroscience: Policies on the Use of Animals and Humans in Neuroscience Research. Approval was

obtained from the IMO and INDEREB Human Participants Ethics committee (reference: CEI/ 029-1/2015), the National Ethics Committee (reference: CONBIOÉTICA-09-CEI-006-20170306), the Research Committee at APEC (17 CI 09 003 142), and the Research Ethics Committee at ENES León (reference: CEI_22_06_S21). Written informed consent was provided by all subjects. All procedures were conducted in accordance with the tenets of the Declaration of Helsinki.

## Animals and models

C57BL/6 mice (male:female ratio = 1) between 6 and 8 weeks of age were obtained from the Institute of Neurobiology's animal house. Much efforts were made to reduce the number of animals at minimum to achieve statistical significance and their suffering (sham manipulation daily a week before testing, use of analgesic drops and semi-invasive ERG electrodes placed at the equator of eyeball, which could not compress eyeball). Male Wistar ($n$ = 20–22) adult (250–300 g) rats ($n$ = 6–8) were used, as well as lean ($n$ = 15) and spontaneous obese ($n$ = 12) *Neotomodon alstoni* mice [36]. Animals were fed *ad libitum* and reared in normal cyclic light conditions (12 h light/dark cycle) with an ambient light level of $\sim$400 lux. Plasma glucose concentrations were measured from a tail blood sample using a blood glucose analyzer (Accucheck active, GC model).

Six- to eight-week-old C57BL/6 mice were divided into two groups of 16 and fed a chow diet (5020, Lab Diets) containing 21% of calories from fat or a high-fat diet containing 60% calories from fat (D12492 Research Diets).

Diabetes was induced in Wistar rats by i.p. injection of streptozotocin (55 mg per kg body weight) in citrate buffer [37]. Control rat group received only citrate buffer injections. We confirmed diabetes by measuring blood glucose (>250 mg/dl in animals [38]) 24 h after streptozotocin injection. Bodyweight was measured as indicated and glycemic controls were always performed after a 6 h fasting [39].

For *in vivo* ERG, animals were evaluated after a 12-h dark adaptation period.

Once experiments were concluded, animals were sacrificed by $CO_2$ inhalation.

## Insulin and glucose tolerance tests

Glucose tolerance tests and insulin tolerance tests were performed on C57BL/6J after 12 weeks of control or high-fat diet and in 1-year-old lean and spontaneously obese *Neotomodon alstoni* mice, after 6-h fasting [39].

The insulin tolerance test consisted in measuring glucose levels with a glucometer in tail vein blood samples obtained with a lancet needle before or 15, 30, 45, 60, and 90 minutes after an ip injection of 1 U/kg insulin (Humulin R; Eli Lilly). For the glucose tolerance test, mice were given glucose at a dose of 2 g/kg through a jugular vein catheter. Blood samples were then collected at 0, 15, 30, 45, 60, and 90 min after glucose administration to measure glycemia. Glucose profiles normalized to the initial glucose reading of each mouse were plotted for each group versus time of subsequent glucose determinations.

## Electroretinograms in animals

Animals were anesthetized with 70% ketamine and 30% xylazine (1 µl/g body weight, ip). Corneas were hydrated with hypromellose (5 mg/mL), tetracaine drops were applied to animal eyes to avoid animal pain, and pupils were dilated with tropicamide-phenylephrine (50 mg/8 mg/mL). ERG responses were recorded with contact lens silver electrodes [40] (3.0 mm diameter, Ocuscience) placed at the equator of eyeball, which could not compress eyeball. Reference electrodes were positioned subcutaneously exactly between the eyes.

The signal was amplified x100; the bandpass was set at 0.1 Hz to 1 kHz (AC-DC Differential Amplifier, Model 3000, A-M Systems) and acquisition frequency to 1 kHz (USB-6009, National Instruments). Spontaneous mesopic activity was measured for 10 minutes after a 20-minute dark adaptation period (0.1 lux) and then spontaneous photopic activity for 10 minutes after adaption to normal light (400 lux) for 10 min. At the end, light stimulation (0.7 ms flashes of 0.38 log cd.s/m$^2$; MGS-2 white Mini-Ganzfeld Stimulator, LKC Technologies) was given to confirm retina function. If no classical evoked response was seen [41], data were discarded.

## Human data description

A total of 520 adult subjects aged between 30 and 80 years (mean: 45.27 ± 0.82 years, 265 females), metabolically healthy or with overweight, obesity, MetS, or diabetes but no DR, were enrolled between February 26, 2015 and December 2019 and from September 2021 and June 17, 2022, in the Mexican Institute of Ophthalmology (IMO) of Querétaro (mean age: 51.39 ± 1.49 years, 27 females), between November 11 and December 20, 2019 and from January 6 and May 26, 2022, in the Instituto de la Retina del Bajío (INDEREB) in Querétaro (mean age: 32.98 ± 2.07 years, 24 females), between August 10, 2021 and March 20, 2022 in the Asociación Para Evitar la Ceguera (APEC) in Mexico city (mean age: 45.77 ± 1.20 years, 119 females), and between August 10, 2021 and May 31, 2022 in the Clínica de Salud Visual (CSV) at ENES León UNAM in León (mean age: 44.96 ± 0.84 years, 96 females). 375 (age mean: 46.01 ± 0.98 years, 180 females) completed all tests required for the current study.

Subjects underwent an anamnesis and an initial optometric examination to ensure that they were eligible to participate. The exclusion criteria were ages outside 30 to 80 range, lens opacity, myopia greater than 6 diopters, glaucoma or other concomitant ophthalmologic disorders, ocular anomalies (e.g., surgery, trauma), recent use of laser or anti-angiogenic intravitreal administration, and cornea problems that disable ERG recordings.

Qualified medical personnel collected the anthropometric data in the morning (8 am to 9 am) after an overnight fast. Height was measured to the nearest 0.5 cm with a stadiometer (Seca 213; Seca). Bodyweight was measured with subjects wearing light clothing and without shoes to the nearest 0.1 kg on a mechanical column scale (Seca 700; Seca). Waist circumference was measured on undressed subjects at the midpoint between the lower margin of the last palpable rib and the top of the iliac crest while the subject was standing, after a moderate expiration, with a non-stretchable tape. Blood pressure was measured by using a mercury cuff sphygmomanometer after the study participant had been quietly seated for ≥10 min. Blood samples were taken from an intravenous catheter without stasis after an overnight fast of at least 8 h. Laboratory measurements that include fasting blood glucose, glycated hemoglobin (HbA1c), insulin, triglycerides (TG), low-density lipoprotein (LDL) cholesterol, very low-density lipoprotein (VLDL) cholesterol, high-density lipoprotein (HDL) cholesterol, total cholesterol (CT), and creatinine were performed at INTERMEDIC (Querétaro, Mexico) for IMO data, iml Laboratorio Médico (Querétaro, Mexico) for INDEREB data, Laboratorio clínico Jenner (Mexico City, Mexico) for APEC data, and Laboratorios Salud Digna (León, Mexico) for CSV data. The homeostasis model of assessment index (HOMA-I) was calculated using fasting insulin and glucose values [42].

All patients were classified according to the following criteria: normoglycemia (fasting glucose < 6.1 mmol/l) and diabetes (fasting glucose ≥ 7.0 mmol/l), according to the 1999/2006 WHO criteria [43, 44]. Normal weight was defined body mass index (BMI) between 18.5 and 24.99 kg/m$^2$, overweight with a BMI between 25 and 29.99 kg/m$^2$, and obesity with a BMI over 30 kg/m$^2$. MetS was defined according to the International Diabetes Foundation criteria for

**Table 1. Characteristics of the patients studied for model training and test using ERG spectral components from three different recording devices.**

| | CONTROL | OVERWEIGHT | OBESITY | MetS | DIABETES without DR | p |
|---|---|---|---|---|---|---|
| n | 80 | 40 | 14 | 66 | 68 | |
| Age (years) | 41.49 ± 1.88[d] | 42.70 ± 2.08[cd] | 39.57 ± 2.93[cd] | 60.45 ± 1.79[bc] | 65.86 ± 1.49[a] | <0.0001 |
| DM1 | - | - | - | - | 7 | |
| DM2 | - | - | - | - | 61 | |
| Duration of diabetes (years) | - | - | - | - | 9.45 ± 1.04[c] | 0.0077 |
| Body weight (Kg) | 58.38 ± 1.23[c] | 72.47 ± 1.40[ab] | 79.12 ± 3.97[a] | 69.65 ± 4.41[a] | 76.48 ± 1.86[a] | <0.0001 |
| Waist circumference (cm) | 83.13 ± 1.30[c] | 96.05 ± 1.38[ab] | 104.14 ± 4.26[a] | 88 ± 3.77[a] | 102.91 ± 1.53[a] | <0.0001 |
| Hip circumference (cm) | 82.28 ± 1.25[c] | 93.80 ± 1.38[b] | 99.46 ± 3.94[ab] | 96.36 ± 3.96[ab] | 103.17 ± 1.18[a] | <0.0001 |
| BMI (Kg/m$^2$) | 22.39 ± 0.31[d] | 26.85 ± 0.25[bc] | 29.93 ± 1.82[abc] | 27.12 ± 1.08[ab] | 30.14 ± 0.63[a] | <0.0001 |
| Glycemia (mg/dl) | 86.77 ± 0.79[b] | 87.75 ± 1.21[b] | 91.43 ± 3.09[b] | 144.82 ± 20.06[b] | 147.48 ± 6.39[a] | <0.0001 |
| HbA1c (%) | 5.30 ± 0.03[d] | 5.47 ± 0.05[d] | 5.39 ± 0.10[d] | 8.41 ± 0.69[d] | 7.67 ± 0.24[c] | <0.0001 |
| Insulinemia (μUI/ml) | 7.36 ± 0.44[c] | 9.54 ± 0.91[c] | 11.15 ± 1.74[abc] | 9.84 ± 1.70[bc] | 21.50 ± 2.89[ab] | <0.0001 |
| HOMA-I | 1.67 ± 0.10[b] | 2.06 ± 0.21[b] | 2.64 ± 0.49[ab] | 3.44 ± 0.70[b] | 8.66 ± 1.31[a] | <0.0001 |
| TG | 100.37 ± 4.79[a] | 104. 70 ± 6.99[c] | 104.94 ± 12.56[bc] | 132.55 ± 21.37[a] | 180.61 ± 11.05[ab] | <0.0001 |
| CT (mg/dl) | 179.55 ± 3.61[b] | 185.12 ± 4.37[ab] | 192.19 ± 8.05[ab] | 190.17 ± 12.66[a] | 183.36 ± 4.39[b] | 0.0015 |
| HDL (mg/dl) | 56.92 ± 2.02[a] | 54.64 ± 1.75[ab] | 56.90 ± 3.36[abc] | 48.84 ± 5.21[c] | 46.13 ± 1.45[c] | <0.0001 |
| LDL (mg/dl) | 105 ± 2.72[b] | 117.27 ± 3.73[ab] | 118.25 ± 7.97[ab] | 113.06 ± 11.02[a] | 108.55 ± 3.84[ab] | 0.00889 |
| VLDL (mg/dl) | 17.23 ± 1.08[d] | 17.62 ± 1.98[d] | 17.05 ± 1.94[cd] | 26.51 ± 4.27[ab] | 30.34 ± 2.37[bc] | <0.0001 |
| Creatinine (mg/dl) | 0.79 ± 0.02[b] | 0.77 ± 0.02[ab] | 0.75 ± 0.04[ab] | 1.09 ± 0.18[ab] | 0.79 ± 0.05[b] | 0.00965 |
| Systolic blood pressure (mmHg) | 114.73 ± 2.23[c] | 114.79 ± 2.82[c] | 124.14 ± 7.75 [bc] | 125.64 ± 4.72[ab] | 131.33 ± 2.62[b] | <0.0001 |
| Diastolic blood pressure (mmHg) | 71.64 ± 1.16[d] | 74.05 ± 1.65[cd] | 73.14 ± 3.21[bcd] | 85.82 ± 4.31[a] | 82.61 ± 1.50[ab] | <0.0001 |

Values, mean ± s.d. *P* values were determined by the Welch ANOVA test. Values that do not share a letter (a, b, or c) are statistically different. Years, y.

MetS [45], i.e. central obesity—BMI>30 kg/m$^2$ or waist circumference > = 94 cm in male and > = 80 cm in female—plus any two of the following four factors: raised TG, reduced HDL, raised blood pressure, raised fasting plasma glucose, raised HbA1c, raised plasma insulin, TG, LDL, VLDL, total cholesterol, creatinine, HOMA-I, or atherogenic index (calculated as $\log_{10}$ (TG/HDL) [46]). Data from IMO, INDEREB, and APEC (*n* = 307) were used for model training and test, and data from CSV (*n* = 39) were used for external validation of the predictive diagnosis model. Tables 1 and 2 contain an overview of patient demographics and biometrics for training/test and external validation of the predictive diagnosis model, respectively.

All subjects underwent a complete ophthalmologic examination including visual acuity testing using Snellen primer; anterior segment and crystalline status under microscopy and indirect ophthalmoscopy with a magnifying glass of 20 diopters; intraocular pressure by flattening tonometry (iCare TA01i); photographic study (7-field color photographs under pupil dilation, ZEISS camera, FM/NA, 60-degree images at IMO; ZEISS clarus® 500 Fundus Camera at INDEREB and CSV; and Visucam® 500 at APEC; and macular patterns, raster and macular thickness map by optical coherence tomography (OPTOVUE RTV-1000 equipment at IMO; Spectralis® Heidelberg Engineering at INDEREB and APEC; and CIRRUS HD-OCT 5000, Zeiss at CSV). Ophthalmic ERG tests were also performed, following ISCEV guidelines [47]. At IMO, the dark-adapted 0.01, 3.0, and 10, dark-adapted 3.0 oscillatory potentials, light-adapted 3.0, light-adapted 3.0 flicker, and multifocal ERG responses were measured in the order indicated using the MonElec2 (Metrovision, France; from February 26, 2015 to December 20, 2017) or Retimax Advanced (CSO, Italy; from January 10, 2018 to June 17, 2022). At INDEREB, APEC, and CSV, the dark-adapted 0.01, 3.0, and 10, light-adapted 3.0, light-adapted 3.0 flicker ERG responses and the DR score [48] were recorded using a mydriasis-free

**Table 2. Characteristics of the patients studied for external model validation.**

| | CONTROL | OVERWEIGHT | OBESITY | MetS | DIABETES without DR | p |
|---|---|---|---|---|---|---|
| n | 19 | 6 | 4 | 5 | 4 | |
| Age (years) | 30.50 ± 3.12[c] | 38.80 ± 7.62[c] | 33 ± 2.12[bc] | 31.5 ± 2.47[c] | 67.33 ± 7.31[a] | <0.0001 |
| DM1 | - | - | - | - | - | |
| DM2 | - | - | - | - | 3 | |
| Duration of diabetes (years) | - | s- | - | - | 2.55 ± 1.73[a] | <0.0001 |
| Body weight (Kg) | 53.57 ± 2.32[b] | 70.98 ± 2.95[a] | 67.55 ± 1.52[ab] | 74.60 ± 3.25[a] | 82.20 ± 1.98[a] | <0.0001 |
| Waist circumference (cm) | 86.13 ± 2.82[b] | 102.52 ± 2.23[a] | 89 ± 0.53[ab] | 101.75 ± 0.53[ab] | 95.80 ± 1.56[ab] | <0.0001 |
| Hip circumference (cm) | 73.56 ± 3.75[c] | 88.94 ± 5.01[bc] | 97 ± 1.41[abc] | 93 ± 1.77[abc] | 105.90 ± 0.64[ab] | <0.0001 |
| BMI (Kg/m$^2$) | 20.38 ± 0.67[c] | 26.40 ± 0.45[ab] | 26.66 ± 0.73[bc] | 28.08 ± 0.48[ab] | 27.97 ± 1.02[ab] | <0.0001 |
| Glycemia (mg/dl) | 84.89 ± 2[b] | 83.60 ± 1.71[b] | 87 ± 5.66[b] | 89 ± 2.83[b] | 118 ± 4.50[b] | <0.0001 |
| HbA1c (%) | 5.32 ±0.03[a] | 5.62 ± 0.12[a] | 5.20 ± 0.10[a] | 5.35 ± 0.32[a] | 7.03 ± 0.38[b] | <0.0001 |
| Insulinemia (μUI/ml) | 7.49 ± 1.25[b] | 10.14 ± 1.84[b] | 15.25 ± 6.26[b] | 14.15 ± 1.38[b] | 18.30 ± 3.60[a] | 0.00245 |
| HOMA-I | 1.61 ± 0.29[b] | 1.84 ± 0.26[b] | 3.45 ± 1.59[b] | 2.85 ± 0.18[b] | 5.40 ± 0.98[a] | 0.0483 |
| TG | 76.75 ± 8.74[b] | 93.84 ± 13.38[b] | 110.55 ± 1.47[b] | 290 ± 46.64[a] | 142.85 ± 17.07[b] | <0.0001 |
| CT (mg/dl) | 177.21 ± 4.26 | 173.13 ± 0.25 | 183.27 ± 0.07 | 205.56 ± 1.02 | 184.13 ± 0.32 | 0.5103 |
| HDL (mg/dl) | 55.69 ± 7.30[b] | 55.74 ± 3.20[b] | 45.49 ± 3.53[a] | 39.16 ± 6.45[a] | 44.36 ± 3.28[a] | 0.0345 |
| LDL (mg/dl) | 98.35 ± 7.66[b] | 112.26 ± 0.18[ab] | 111.94 ± 0.09[ab] | 123.27 ± 0.61[ab] | 102.60 ± 0.32[ab] | 0.04646 |
| VLDL (mg/dl) | 9.36 ± 1.57[bc] | 12.94 ± 2.52[bc] | 14.75 ± 0.51[bc] | 45.10 ± 7.77[a] | 33.33 ± 2.85[b] | <0.0001 |
| Creatinine (mg/dl) | 0.78 ± 0.04[b] | 0.77 ± 0.04[b] | 0.78 ± 0.02[b] | 0.80 ± 0.04[ab] | 0.91 ± 0.08[ab] | 0.00418 |
| Systolic blood pressure (mmHg) | 106.89 ± 3.71 | 109 ± 6.41 | 110.50 ± 8.13 | 138.50 ± 3.18 | 131 ± 1.41 | 0.0643 |
| Diastolic blood pressure (mmHg) | 73.33 ± 3.43 | 76.80 ± 4.47 | 66 ± 6.36 | 78 ± 3.54 | 86 ± 9.19 | 0.059 |

Values, mean ± s.d. P values were determined by the Welch ANOVA test. Values that do not share a letter (a, b, or c) are statistically different. Years, y.

ERG device (RETeval complete, LKC Technologies, USA). OP features (N1, P1, and N2 amplitudes and peak times, N1-P1 and P1-N2 ratios) were extracted from ISCEV dark-adapted 3.0 ERG protocol [47]. The DR score derived from flicker ERG and pupillography data correlate with ocular intervention for DR [48]. In addition, spontaneous ERG responses were measured in all patients using a custom protocol developed specifically for each ERG device (no flashlight, 0.3–1000 Hz band-pass filter with a 50 Hz notch, 1 kHz acquisition, and ×100,000 gain). At IMO, 30 min prior to ERG recording, one drop of tropicamide 1% was instilled into each eye as a cycloplegic and contact lens electrodes were used posterior to corneal anesthesia with proparacaine hydrochloride eye drops. At INDEREB, APEC and CSV, skin electrodes on the lower eyelid were used [49]. All patients were adapted to mesopic conditions for 20 minutes prior dark-adapted ERG assessment and before ERG assessment under photopic conditions, patients were adapted to normal light (400 lux) for 10 min. The acquisition sequence was as follows: spontaneous dark-adapted ERG, evoked dark-adapted ERGs, spontaneous light-adapted ERG, and evoked light-adapted ERGs. Electrode impedance was monitored throughout the test and maintained below 10 KOhm by repositioning the electrodes as required. Using an inbuilt artefact rejection algorithm, the electrophysiology software automatically detected artefacts (e.g., from blinking), removed the corresponding responses and retested the sequence. Imaging, ERG, and optometric measurements were performed by certified technicians or ophthalmologists.

Patient diagnosis for DR or other eye issues was established once by experts at IMO (M.G. R.), INDEREB (R.G.F.), APEC (H.Q.), and CSV (L.F.H.Z.) and after the finalization of our predictive diagnosis model, a second diagnosis (E.L.S. and V.C.S. from IMO) was established to compare our model performance with that of experts. The second diagnosis by experts was

**Table 3. Characteristics of the patients studied for external model validation using ERG spectral components from one recording device.**

| | CONTROL | OVERWEIGHT | OBESITY | MetS | DIABETES without DR | p |
|---|---|---|---|---|---|---|
| n | 37 | 18 | 8 | 42 | 33 | |
| Age (years) | 34.92 ± 2.85[c] | 38.83 ± 3.59[bc] | 38.25 ± 3.12[bc] | 46.12 ± 2.02[b] | 60.58 ± 2.05[a] | <0.0001 |
| DM1 | - | - | - | - | 1 | |
| DM2 | - | - | - | - | 32 | |
| Duration of diabetes (years) | - | - | - | - | 6.15 ± 1.12[b] | 0.0003 |
| Body weight (Kg) | 55.03 ± 1.45[c] | 71.41 ± 1.80[ab] | 75.34 ± 6.62[ab] | 78.20 ± 2.03[a] | 72.19 ± 1.99[ab] | <0.0001 |
| Waist circumference (cm) | 83.24 ± 3.05[b] | 96.95 ± 1.80[a] | 97.81 ± 4.65[a] | 102.63 ± 1.30[a] | 102.25 ± 1.74[a] | <0.0001 |
| Hip circumference (cm) | 73.82 ± 2.94[b] | 86.69 ± 5.64[ab] | 96.28 ± 5.85[a] | 98.92 ± 1.78[a] | 101.69 ± 1.41[a] | <0.0001 |
| BMI (Kg/m$^2$) | 21.15 ± 0.35[c] | 26.78 ± 0.34[ab] | 26.48 ± 1.91[ab] | 29.03 ± 0.65[a] | 29.71 ± 0.73[a] | <0.0001 |
| Glycemia (mg/dl) | 82.84 ± 1.08[c] | 85.18 ± 1.14[c] | 90.50 ± 1.85[c] | 89.93 ± 1.87[c] | 132.42 ± 8.56[b] | <0.0001 |
| HbA1c (%) | 5.32 ± 0.04[d] | 5.48 ± 0.06[d] | 5.31 ± 0.08[d] | 5.62 ± 0.06[d] | 7.42 ± 0.30[c] | <0.0001 |
| Insulinemia (μUI/ml) | 7.82 ± 0.64[b] | 9.99 ± 1.35[b] | 10.89 ± 1.92[b] | 13.65 ± 1.29[ab] | 18.22 ± 4.15[ab] | 0.001 |
| HOMA-I | 1.61 ± 0.12[b] | 2.02 ± 0.27[b] | 2.46 ± 0.47[b] | 3.11 ± 0.33[b] | 5.69 ± 1.12[ab] | <0.0001 |
| TG | 107.41 ± 8.26[b] | 96.89 ± 8.67[b] | 132.63 ± 18.93[b] | 210.66 ± 18.48[b] | 188.76 ± 13.18[b] | <0.0001 |
| CT (mg/dl) | 176.20 ± 4.53[b] | 171.70 ± 7.88[b] | 186.57 ± 11.43[ab] | 201.50 ± 5.56[ab] | 182.93 ± 6.09[b] | 0.012 |
| HDL (mg/dl) | 54.29 ± 2.78[a] | 53.09 ± 2.95[a] | 54.96 ± 5.99[a] | 44.62 ± 1.41[a] | 46.83 ± 1.58[a] | 0.028 |
| LDL (mg/dl) | 106.62 ± 3.70[a] | 113.24 ± 6.35[a] | 112.39 ± 11.96[a] | 124.43 ± 4.70[a] | 106.35 ± 5.33[a] | 0.128 |
| VLDL (mg/dl) | 14.84 ± 1.42[c] | 13.11 ± 1.50[c] | 19.15 ± 3.22[bc] | 32.02 ± 2.85[ab] | 30.19 ± 2.77[b] | <0.0001 |
| Creatinine (mg/dl) | 0.81 ± 0.03[b] | 0.82 ± 0.03[b] | 0.81 ± 0.04[b] | 0.90 ± 0.08[b] | 0.84 ± 0.07[b] | 0.001 |
| Systolic blood pressure (mmHg) | 113.73 ± 3.11[b] | 119.17 ± 3.43[ab] | 109.13 ± 3.59[b] | 133.93 ± 2.75[a] | 135.00 ± 3.87[a] | <0.0001 |
| Diastolic blood pressure (mmHg) | 71.79 ± 1.81[b] | 76.89 ± 1.58[ab] | 70.75 ± 3.25[b] | 85.19 ± 1.25[a] | 80.97 ± 2.23[a] | <0.0001 |

Values, mean ± s.d. *P* values were determined by the Welch ANOVA test. Values that do not share a letter (a, b, or c) are statistically different. Years, y.

made under three conditions, the first with access to full eye examinations, the second with access to full eye tests and reference blood tests for diabetes, and the last with access to full eye tests, full blood tests, and anamnesis.

To test if ERG data from one sensor could improve of our model performance, a group of patients tested with the mydriasis-free RETeval ERG device was enrolled at APEC, INDEREB, and CSV from January to June 2022. Table 3 contains an overview of these patient demographics and biometrics.

## ERG data processing

For spectral analysis of both human and animal ERGs, signals were initially low-pass filtered at 1 kHz and high-pass filtered at 0.1 and 0.3 Hz for animals and humans, respectively. Recordings with large artifacts (which exceeded ± 100 μV) were removed. Recordings from the two eyes were independently analyzed. Raw ERG signals were normalized between -1 and +1.

ERGs show discontinuous activity, reason for which the wavelet (Morlet) transform was used to analyze them. Analysis was carried out with the MATLAB-based fieldtrip toolbox implementing the wavelet method [50]. The data were analyzed using custom-made MATLAB scripts (MATLAB R2018; MathWorks). Spontaneous human ERG signals were transformed within consecutive epochs of 10, 30, and 60 s. The corresponding number of windows was 30, 10, and 5, respectively. Time and spectral resolutions were 0.01 s and 0.05 Hz, respectively. The wavelet transform data were represented as scalograms or normalized power spectra obtained by averaging the wavelet transform throughout the recording. The latter were subsequently grand averaged across all samples, animals, and patients for each condition. The standard error of the mean of the power spectra was calculated across animals/subjects.

In exploratory analysis, frequency points were initially considered between 0.1 or 0.3 Hz and 1 kHz (not shown) in animals and humans, respectively, and then refined in ranges where activity was detected (0.1–10 Hz and 0.3–40 Hz for animals and humans, respectively).

Oscillatory potentials (OPs) were digitally isolated from the scotopic B-wave using a 100–500 Hz digital filter.

## Modeling structure and development

Four common classification algorithms with built-in feature selection (Random Forest, deep learning and linear and radial support vector machines) were applied on human datasets using the open-source R package caret (version 6.0–73) for support vector machines and the $H_2O$ package for Random Forest and deep learning. Random Forest was applied on animal datasets using the $H_2O$ package. In all cases, the final datasets were randomly divided into training (80% of observations) and testing (20%) sets. Only validation dataset results were reported. Random Forest parameters were tuned as follows: ntrees = c(50, 70, 90, 100, 150, 200, 250, 300, 350, 400, 450, 500), max_depth = c(9, 10, 11, 12, 13, 14, 15, 16, 17, 19, 20),min_rows = c (1,2,3,4). We used the option in caret (precision, accuracy, sensitivity, and specificity) and $H_2O$ (ROC, ROC AUC, and confusion matrix) to return class probabilities for all classifiers.

## Model performance analysis

Classes were balanced for all predictions. We therefore computed test performance metrics, including ROC curves [51, 52], AUC-ROC, accuracy, sensitivity, specificity, precision, negative predictive value (NPV) [53], and confusion matrix [54]. The performance of our model was contrasted to the second patient diagnosis by experts using precision, recall, and F1-score.

## Statistical analysis

Statistical analyses were performed using Matlab (Statistics and Machine Learning Toolbox). Data are reported as mean ± s.e.m. or ± 95% confidence interval. All data showed normal distribution and equal variance according to the D'Agostino–Pearson omnibus and Levene tests, respectively. Statistical significance was therefore determined either using unpaired t-test or, for multiple comparisons, using a mixed ANOVA and Bonferroni post-hoc. Human metrics were analyzed using the Welch ANOVA that accounts for variance heterogeneity.

For explanatory statistical analyses, the following variables were considered: BMI, hip and waist circumferences, TG, HDL, blood pressure, fasting plasma glycemia, HbA1c, plasma insulin, LDL, VLDL, CT, creatinine, HOMA-I, atherogenic index, normalized power of 0.3 to 40 Hz oscillations with a frequency resolution of 0.05 Hz, and peak frequencies in the 0.3–2, 10–20, and 20–40 Hz bands. The variables with higher variance (CoV function in R) were selected, i.e., TG, CT, fasting glycemia, LDL, systolic blood pressure, age, body weight, hip circumference, HDL, VLDL, normalized power of 0.45, 0.4, 0.5, 0.55, 0.6 Hz activities, AUC of the 0.3–40 Hz band, normalized power of 0.65, 0.35, 0.7, 0.3, 0.75, 0.8 Hz activities, and AUC of the 20–40 Hz band, (ordered in descending order of variance) to perform PCA analysis. PCA panel and biplots of PC1, PC2, and PC3 were generated in R (princomp, biplot). Linear discriminant analyses were performed in R (library mass; code available in the provided Github access). Cases with missing values were omitted.

To calculate the correlation between the spontaneous oscillation (SO) frequency components of major variance (detailed above), we computed a PCA score for each patient by combining linearly each variable coefficient (PC1) multiplied by its real value. Correlation analysis was done by calculating the Pearson's R coefficient (cor function in R).

The F1 score was calculated to rank our models for the shortest ERG window.

### Code availability

We made use of several open-source libraries to conduct our experiments: *caret* (https://topepo.github.io/caret/) and $H_2O$ (http://docs.h2o.ai/), which provide implementations of individual model components. To facilitate improved reproducibility of our data analyses, the R code and documentation for the analysis are available online (https://github.com/airetinopathydx/AIRetinopathyDx_).

## Results

### Predictive model for obesity, type 1 diabetes, and type 1 diabetes evolution based on spontaneous ERG oscillations in rodents

Spontaneous ERGs of high-fat diet-fed mice that are obese and insulin-resistant after 12 weeks (S1A Fig) showed three consistent peaks in the 0.1–10 Hz range (Fig 1A) of similar power compared to lean mouse ERGs (Fig 1B). The mid-low (1–1.8 Hz) peak frequency was reduced in obese mice, while the low (0.1–0.8 Hz) and mid (2–4 Hz) peak frequencies were not affected (Fig 1B). Spontaneously obese insulin-resistant (S1B Fig) mice and streptozotocin-treated hyperglycemic (S1C Fig) rats exhibited low to mid-low frequency oscillatory activities (Fig 1C and 1E, respectively) of similar power compared to their respective controls (Fig 1D and 1F). The low (0.6–1 Hz) peak frequency (Fig 1D) and the very low (0.2–0.6 Hz) and low-to-mid (0.6–2.5 Hz) peak frequencies (Fig 1F) were reduced in ERGs of spontaneously obese mice and streptozotocin-treated rat, respectively. In contrast, oscillatory potentials (OPs), considered as the most precociously altered ERG parameter in diabetes [55], remain unchanged in high-fat diet-fed mice (S1D and S1E Fig).

We next tested the potential predictive content of these signals measured under the spontaneous modality. A single predictive model based on random forest algorithm was developed using the power spectra of spontaneous ERG oscillations in the activity range between 0.1–10 Hz (see Methods for more details). The area under the receiver operating characteristic (ROC) curve (AUC-ROC) values are 0.804, 0.875, and 0.906 for the high-fat diet-induced obesity mouse model (Fig 1G), the spontaneously obese mouse model (Fig 1H), and streptozotocin-induced type 1 diabetes rat model (Fig 1I), respectively. Additional metrics including accuracy and precision indicate that our model predicts correctly in 80 to 87.5% of cases and is sure of its prediction in 80 to 84.6% of cases (Fig 1G–1I). The specificity of this model is good (>0.75), meaning that it misses few non-diseased cases, and it is extremely sensitive in predicting both obesity and type 1 diabetes cases (Fig 1H and 1J).

Predicting a disease condition is useful for diagnosis at a given disease stage, but diabetes and its complications evolve [33, 56]. Progressive diabetic retinal neurodegeneration has been reported between 2 and 32 weeks of age in rodent models of diabetes [57]. We therefore asked if our model could discriminate the disease cases from control ones over time by using the power spectra of rat ERGs after 4, 6, 8 or 12 weeks of streptozotocin treatment. The performance of this multiclass classification is visualized in a confusion matrix (Fig 1J). Our model correctly identified the week classes in 75% of cases (Fig 1J). These data show that the major earliest risks of DR, i.e., obesity and type 1 diabetes, can be predicted by spontaneous ERG oscillations in rodents.

### Exceeding expert performance to predict preventable risks of type 2 diabetes and thereby DR

To assess the clinical relevance of our approach, we created a human database of spontaneous ERGs (Fig 2A) from 80 metabolically healthy adult subjects (54 eyes), 40 patients with

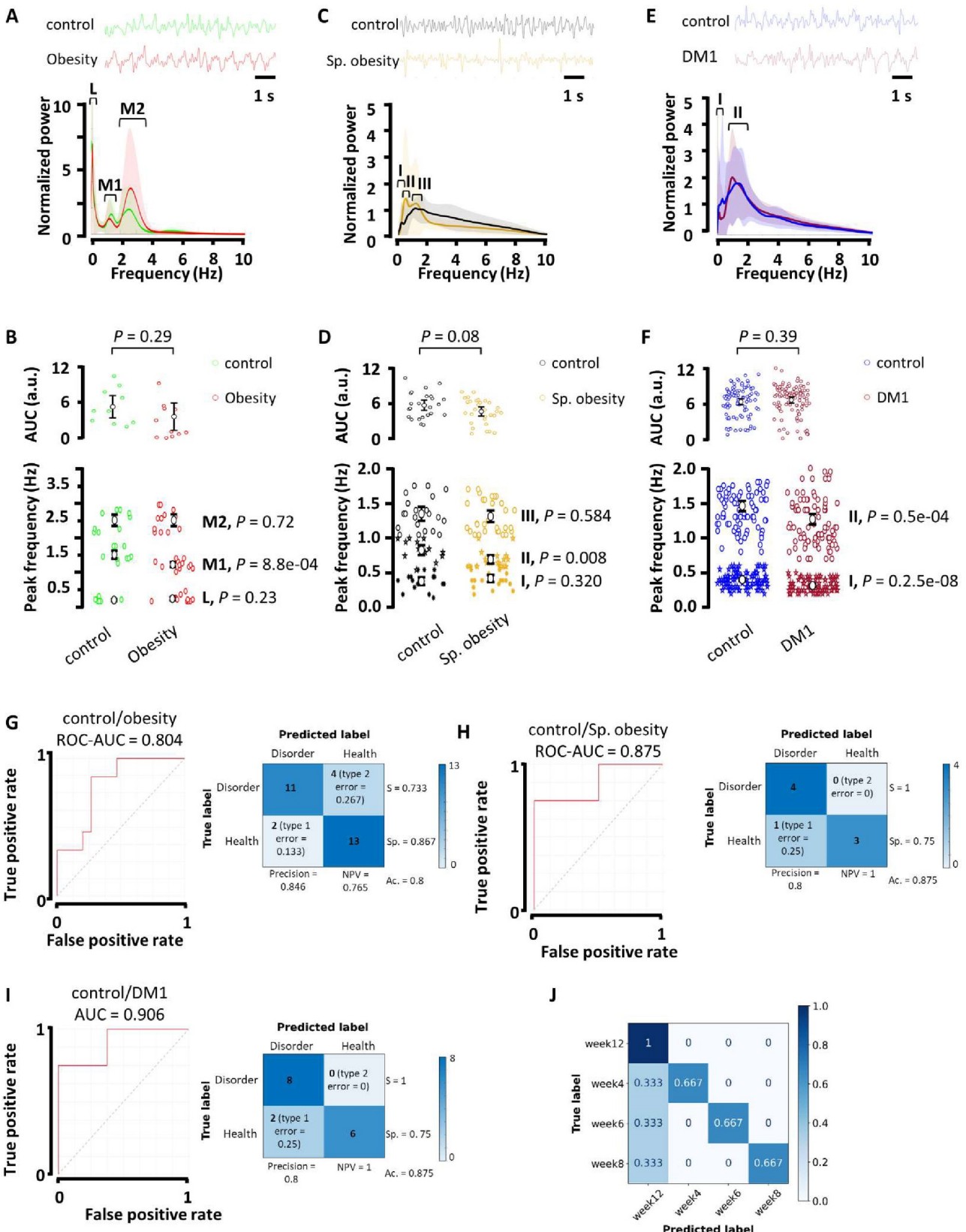

**Fig 1. Spontaneous ERG-based random forest model discriminates control and disease cases in rodent models of obesity and type 1 diabetes and predicts disease evolution in the type 1 diabetes model.** Illustrative spontaneous ERGs and wavelet analysis in A, control versus high-fat diet-fed mice (*n* = 75 and *n* = 75, respectively), C, lean versus spontaneously obese *Neotomodon alstoni* mice (*n* = 20 and *n* = 20, respectively), and E, control and streptozotocin-treated rats (*n* = 40 and *n* = 40, respectively) in the 0.1–10 Hz range, under photopic conditions. ERG signals were normalized. Graphs show the average scalogram power ± s.e.m. throughout 1-minute recordings. The square brackets with the Roman numerals indicate the consistent peaks observed in each control condition. In the high-fat diet-induced obesity model, 0.1–0.8 (Low, L), 1–1.8 (M1, mid-low), and 2–4 Hz (M2, mid) bands were considered; 0.1–0.6 (I), 0.6–1 (II), and 1–1.7 (III) Hz bands in the spontaneous model of obesity, and 0.2–0.6 (I) and 0.6–2.5 (II) Hz bands in the type 1 diabetes model. AUC (0.1–10 Hz) and peak frequency analysis (P values were determined by unpaired Student's t-test) of wavelet graphs in B, the high-fat diet-induced obesity D, the spontaneous model of obesity, and F, the type 1 diabetes models. Graphs show mean ± confidence interval. ROC curves and confusion matrix with performance measures for binary predictions (control vs. experimental cases) using the 0.1–10 Hz power spectra of G, control and high-fat diet-fed mice (*n* = 15 and *n* = 15, respectively), H, lean and spontaneously obese *Neotomodon alstoni* mice (*n* = 4 and *n* = 4, respectively), and I, control and streptozotocin-treated rats (*n* = 8 and *n* = 8, respectively). S, sensitivity; Sp., Specificity; NPV, negative predictive value; Ac., accuracy. J, Confusion matrix of multiclass prediction for the machine learning algorithm that discriminates between control and diseased rats at week 4, 6, 8, or 12 post-streptozotocin injection. Each column represents the instances in a predicted class, and rows represent the instances in an actual class. We used controls at week 4, 6, 8, and 12 (*n* = 8 at each time point).

overweight (24 eyes), 14 patients with obesity (8 eyes), 66 patients with metabolic syndrome but no diabetes (MetS, 56 eyes), and 68 patients with diabetes but no DR (44 eyes) (Table 1), who were referred to both IMO and INDEREB ophthalmology clinics for general eye check-up. Participants with MetS were older than controls (median 60.45 years vs 41.49 years, p<0.001) and patients with diabetes were older than all other groups (median 65.86 years vs 46.05 years, p<0.001), and more likely to be women (79.41% vs 20.58%, *P* < 0.0001) (Table 1). A ground-truth label for the presence of metabolic health, overweight, obesity, MetS, diabetes mellitus, and no DR was established using internationally accepted criteria, as referenced in the Methods. Importantly, OP parameters of the overweight, obese, and MetS groups remain unchanged compared with the control group and overall, they also showed no difference in the diabetes without DR group (S2A–S2C Fig). The DR score [48] is similar between the control, overweight, obese, and MetS group, and is reduced patients with diabetes but no DR (S2D Fig). We further showed that the spontaneous ERG oscillation component was neither correlated with OP parameters (S2E and S2F Fig) nor DR score (S2G Fig).

Spontaneous activity is seen between 0.3 and 40 Hz (Fig 2B), with main peaks in the 0.3–2, 10–20, and 20–40 Hz bands (Fig 2B, insets). The large variation in our dataset that we assumed was confirmed by the absence of statistical differences in AUC and peak frequency between all groups in the previously mentioned bands (Fig 2B). We therefore pooled spontaneous ERG-power spectra from non-healthy participants as one disease group (Fig 2C) and found an increased peak frequency in the 20–40 Hz band in the disease group (Fig 2D). No significant difference was observed in the AUC or peak frequency in the 0.3–2 and 10–20 Hz bands (Fig 2D). Next, the power spectra from 986 aleatory fragments of one-minute duration extracted from the 186 ERGs registered from a total of 268 patients (186 eyes), were randomized across training (80%) and test (20%) sets; diseases cases were defined as all cases except metabolically healthy and no DR, and the same random forest-based model used in animal models was applied in our database of human ERG power spectra.

The predictive model performed good (AUC-ROC = 0.761, Fig 2E) and is accurate in 67.3% of the cases (Fig 2F). The prediction precision is poor and the model shows 85.4% sensitivity and a specificity of 61.5% (Fig 2F). We then aimed to assess the accuracy of our model using an external cohort. To this end, we created a validation database based on data from patients enrolled in an important eye care setting in Mexico City (Table 2). Using this validation dataset of 0.3–40 Hz power spectra (126 aleatory fragments of one-minute duration extracted from the 38 ERGs registered from a total of 39 patients (11 eyes from controls, 5 eyes from overweight, 3 eyes from obese, 3 eyes from MetS, and 3 eyes from diabetes but no DR), the predictive model performance was similar to the one of the original model (Fig 2G and 2H). These data provide an external validation of our predictive model for early DR risks.

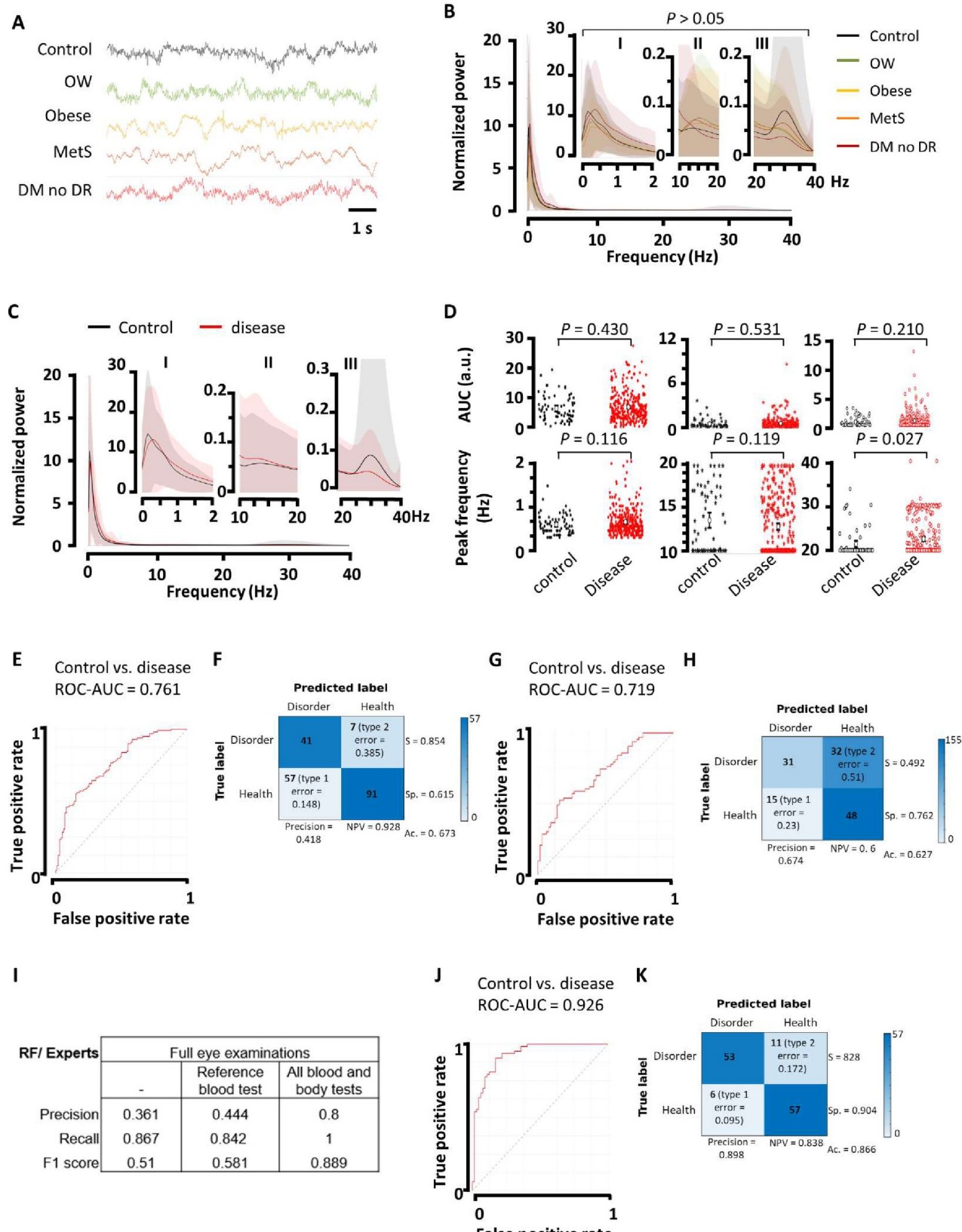

**Fig 2. Spontaneous ERG oscillations help predict the disease in patients with early risk factors of DR.** A, Illustrative signals from spontaneous ERGs of control subjects and patients with overweight (OW), obesity, metabolic syndrome (MetS), and diabetes but no DR (DM no DR) under photopic conditions. Signals are normalized. Wavelet analysis for B, each group separately and C, combining all unhealthy conditions into one disease group. Control group is defined in Methods. Graphs show the average scalogram power ± s.e.m. throughout 20-second recordings between 0.3–40 Hz (host graph) and 0.3–2 (I), 10–20 (II), and 20–40 (III) Hz (inset graphs), where consistent peaks were observed in the control group. D, AUC and peak frequency analysis in the I, II, and III bands of wavelet power spectra from control and disease groups (*P* values were determined by unpaired Student's t-test). Graphs show mean ± confidence interval. E, ROC curve and F, confusion matrix with performance measures for the Random Forest model discriminating control from disease cases in control and disease groups, using the 0.3–40 Hz power spectra. G, ROC curve and H, confusion matrix with performance measures corresponding to the validation of our predictive model thanks to an external and independent validation dataset. I, Classification performance of our model versus experts, when experts have access to full eye examination alone or combined with reference tests to diagnose diabetes (fasting glycemia, creatinine, triglycerides, and cholesterol) or full laboratory (glycemia, HbA1c levels, HOMA-I, and lipid profile) and non-laboratory (blood pressure, weight and waist circumference, and BMI) tests. Precision, recall, and F1-score are reported. I, ROC curve and J, confusion matrix with performance measures corresponding to the performance of our predictive model using ERG-derived power spectra recorded by only one ERG device. In A-F and I, control (metabolically healthy, *n* = 80) and disease (OW, *n* = 40; obese, *n* = 14; MetS, *n* = 66; and diabetes with no DR, *n* = 68) groups. In G, H, control (metabolically healthy, *n* = 11) and disease (OW, *n* = 5; obese, *n* = 3; MetS, *n* = 3; and diabetes with no DR, *n* = 3) groups. In J, K, control (metabolically healthy, *n* = 37) and disease (OW, *n* = 18; obese, *n* = 8; MetS, *n* = 42; and diabetes with no DR, *n* = 33) groups. S, sensitivity; Sp., Specificity; NPV, negative predictive value; Ac., accuracy.

Interestingly, when we fed our model with the power spectra from ERGs registered with one and same device (762 aleatory fragments of one-minute duration extracted from 173 ERGs registered from a total of 237 patients (Table 3, 173 eyes)), the predictive model performed greatly (AUC-ROC = 0.926, Fig 2J) and is accurate in 86.6% of the cases (Fig 2K). The prediction precision is 89.8%; the model shows 82.8% sensitivity and a specificity of 90.4% (Fig 2K).

During the model implementation, we found that the random forest-based model outperformed other models based on other algorithms, including linear and radial support vector machines (S3A and S3B Fig) and deep learning (not shown). We also sought the best performance for the shortest ERG window and lighting conditions to optimize setting conditions. The greatest prediction performance was established for spontaneous photopic ERGs of 60 s (S3C–S3E Fig).

Case labeling relies on expert's knowledge. When we compared our model predictions with the predictions made by medical experts, our model underperformed in terms of precision (0.361) but was consistent in recall (0.867) (Fig 2I). This, when experts only access full eye examinations (Fig 2I). Our model performance slightly improved (precision 0.444) in case experts had access to reference tests to diagnose potential diabetes (Fig 2I). Our model was close to the judgements of experts in ophthalmology if, in addition to eye examination, they had access to laboratory (glycemia, HbA1c levels, HOMA-I, and lipid profile) and non-laboratory (blood pressure, weight and waist circumference, and body mass index (BMI)) tests (Fig 2I). In the three contexts, our model showed a good recall, referring to few false negatives –cases our model would classify as healthy that are not– (Fig 2I). The fact that our model recognized disease cases with less information than experts prompted us to look closely at cases of errors in our model in all three contexts (only complete eye exam, the latter plus reference blood test; the latter two plus specific tests for metabolic profile). It appeared that error cases are false positives (low precision), indicating that our model "incorrectly" labeled cases as positive (or diseased) that were labeled as negative by experts. But are these negative labels from experts truly negative? Experts classified them as "with no retinopathy" based on eye examination, while deeper blood tests revealed that these "negative cases" present risk factors for DR. These data demonstrate that our model can predict early risk factors for DR with less information than experts, outperforming them.

## Statistical explanation for the spontaneous ERG-based model of prediction for preventable risks of type 2 diabetes and thereby DR

When we implemented a multiclass model encompassing all risk stages for diabetes and DR, classification performance reached 66% (Fig 3A). The largest errors of our model consisted in

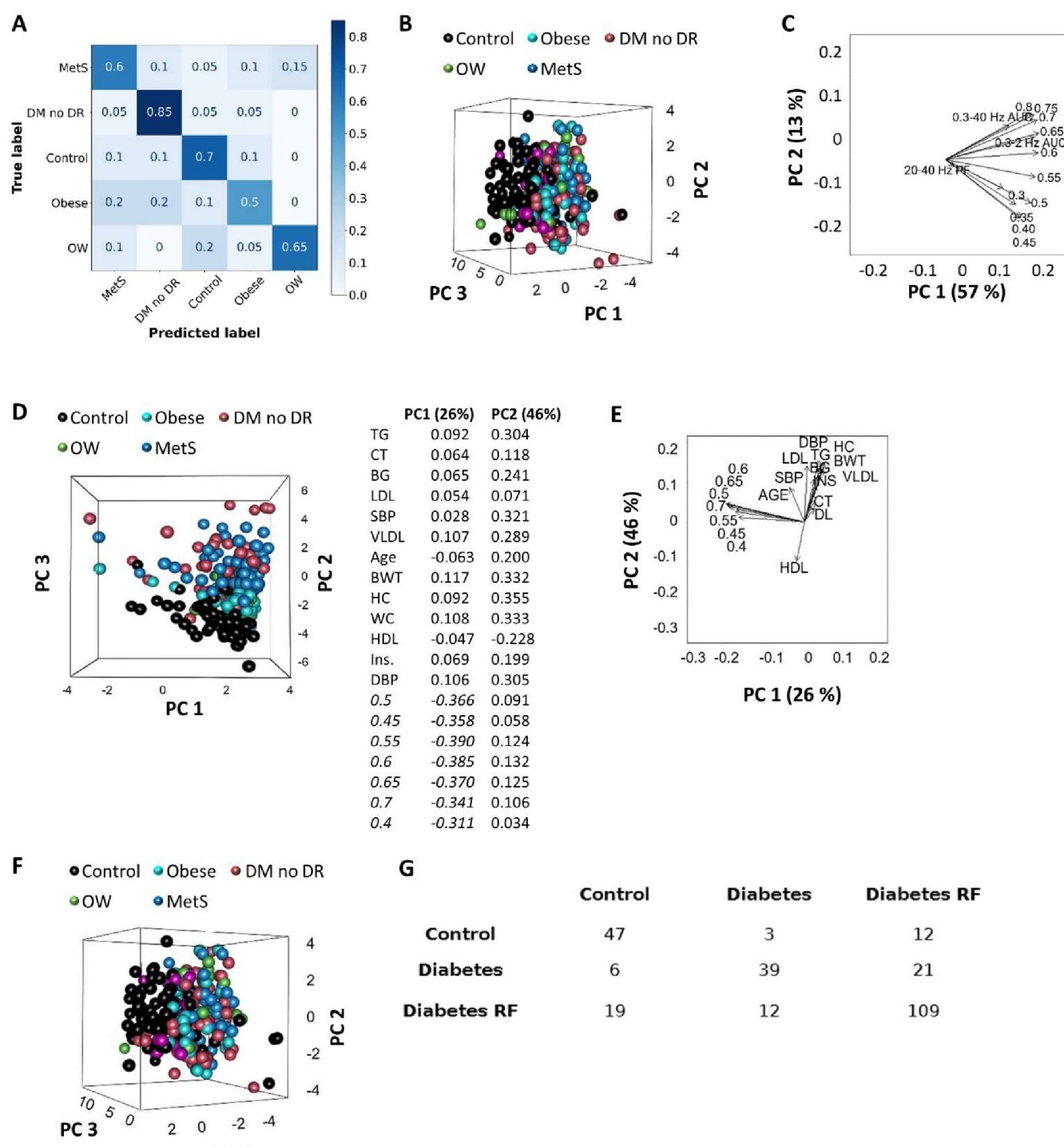

**Fig 3. Multiclass prediction of people at risk of developing type 2 diabetes and DR using the spontaneous ERG-based random forest model: Explanatory data analysis.** A, Multiclass confusion matrix for the prediction of control, overweight (OW), obesity, MetS, and diabetes with no DR (DM no DR) once the first binary classification (control vs. disease) was done using the Random Forest model. Data from Tables 1 and 2 were pooled and aleatory divided in a training (80%) and test (20%) set. B, Separation of control, overweight, obese, metabolic syndrome, and diabetes (DM) without DR cases using PCA and C, corresponding biplot of PC1 and PC2, using 13 variables: normalized power of the slowest oscillations (0.3–0.8 Hz), the AUC of the 0.3–40 Hz band (0.3–40 Hz AUC), and the peak frequency of the 20–40 Hz band (20–40 Hz PF). D, Separation of control, overweight, obese, metabolic syndrome, and diabetes (DM) without DR cases using PCA and E, corresponding biplot of PC1 and PC2, using 20 variables: normalized power of 0.4–0.7 Hz activities, TG, CT, fasting blood glucose (BG), LDL, systolic blood pressure (SBP), VLDL, age, body weight (BWT), hip and waist circumferences (HC and WC, respectively), HDL, insulinemia (Ins.), and diastolic blood pressure (DBP). Table (right) shows variable contributions to the first and second principal

components. F, Separation of control, overweight, obese, metabolic syndrome, and diabetes (DM) without DR cases using PCA, using 8 variables: normalized power of 0.7 to 0.8 Hz activities, the peak frequency in the 20–40 Hz band, the AUC of the 0.3–40 Hz band, age, BG, CT. G, Confusion matrix corresponding to the linear discriminant analysis to separate control, diabetes risk factors (overweight, obesity, and MetS, n = 168), and diabetes (DM) without DR groups. From B to G, control (n = 62), overweight (OW, n = 66), obese (n = 17), metabolic syndrome (MetS, n = 85), and diabetes (DM) without DR (n = 41).

classifying obesity into more advanced states and some overweight cases into controls (Fig 3A).

To gain insights into the discrimination between groups, we performed explanatory statistical analyses. We considered the variables with higher variability (detailed in Methods), including the normalized power of the slowest oscillations (0.3–0.8 Hz), the AUC of the 0.3–40 Hz band, and the peak frequency of the 20–40 Hz band. Measurements of the considered plasma and body variables significantly differed between control and disease groups (Tables 1 and 2), no significant change in the power of slow frequency spontaneous oscillations of the ERG was detected, the peak frequency in the 20–40 Hz band increased in the disease group (Fig 2B and 2D). The initial evaluation of the utility of these variables in separating control and disease groups was accomplished by examining their natural partitioning using Principal Component Analysis (PCA). When only using variables related to spontaneous ERG oscillations, no separation of all groups was observed, but control cases tend to cluster (Fig 3B). PC1 and PC2 explained 69% of the data set variance, powers of 0.6–0.7 Hz oscillations correlate with the AUC of the 0.3–2 and 0.3–40 Hz bands and had the largest contributions to PC1 (Fig 3C). The separation of the control group could be better appreciated when spontaneous ERG oscillation-related variables were combined with body variables like triglyceridemia, total cholesterol levels, fasting blood glucose, very low-, low-, and high-density lipoprotein cholesterol, systolic and diastolic blood pressure, age, body weight, hip and waist circumferences, and insulinemia. PC1 and PC2 explained 47% of the data set variance (Fig 3D). According to our PCA results, the normalized power of the slowest oscillations (from 0.4 to 0.7 Hz) was the most important in separating disease cases from controls as they had the largest contributions to PC1 (Fig 3D). Plasma triglycerides, total cholesterol, low- and high-density lipoprotein cholesterol, fasting blood glucose, insulin, systolic blood pressure, age, and hip circumference, were the least important in this regard (Fig 3D). Based on visual inspection of the PC1/PC2 biplot (Fig 3E), we refined our PCA analysis to variables related to spontaneous ERG oscillations (from 0.7 to 0.8 Hz, the peak frequency in the 20–40 Hz band, and the AUC of the 0.3–40 Hz band) that were non-orthogonal with body-related variables (age, fasting blood glucose levels, and total cholesterol), but PCA did not show a better separation of disease and control cases (Fig 3F). We finally used linear discriminant analysis since it is geared towards the discrimination between user-defined groups [58]. As shown in the confusion matrix of the best performances we found for discriminating control (76%), risk factors for diabetes (78%), and diabetes (60%) cases (Fig 3G), this relative success required the combination of body metrics in addition to spontaneous ERG oscillation-related variables.

## Discussion

The clinical community is experiencing an explosion of machine learning-guided (tele)diagnostics, particularly in the context of detecting treatment-requiring DR, and the positive impact on care delivery is becoming evident [59, 60]. Risk-scoring algorithms for undiagnosed diabetes are also available [61], but there is no such approach for the main risk factors of type 2 diabetes, i.e., overweight, obesity, and MetS with not yet diabetes. The purpose of screening such conditions resides in well-established data about their preventability with adopting healthy lifestyle behaviours [62], and because people of normal weight are not always

metabolically healthy [63] and at the opposite, the obesity phenotype can associate with no or little evidence of metabolic dysfunction [64]. We have exploited the potential of non-invasive, quantitative, and objective ERG by introducing a new and simple modality to record spontaneous activity of the retina, and combined it with a supervised machine learning-based model to predict early risk factors of type 2 diabetes, and thereby DR. Our data support the robustness of the random forest system in the screening of early risk factors for type 2 diabetes, and imply its translation into clinical use thanks to automated platforms like the one we created (http://deepretinopathydx.inb.unam.mx/). Our findings also highlight a very early impact of systemic metabolic changes in spontaneous signals from the central nervous system and add to the growing list of evidence that show retinal neurodegeneration as an early event in DR pathogenesis [65].

## Strengths and interpretation of spontaneous ERG signals' predictive content

We found that the spontaneous oscillations detected by non-evoked ERG are a quantifiable biological parameter that is modified under conditions related to an excess of body weight. These alterations were not accompanied by changes in OPs in high-fat diet-fed mice and in people with overweight, obesity, and MetS. These observations are consistent with previous studies in animal models [66, 67], but contrast with data showing reduced OP amplitude in high-fat diet-fed mice [68] or in ob/ob mice [69]. These discrepancies may be attributed to the different induction mechanisms of obesity, known to result in models with their own characteristics [38, 70]. Particularly, our high-fat diet is more enriched in lipids than carbohydrates [68]. While the ERG B-wave is affected in patients with obesity [71, 72], there is no previous report of the effect of obesity on OPs in humans. In patients with diabetes but no DR, the amplitude and implicit time of the OPs tend to decrease and increase, respectively, as previously reported [73]. In general, these data agree with a view that the spontaneous activity of the retina may replace OPs [55], by becoming the most precociously altered functional parameter in stages previous to diabetes. In further agreement with this, is our finding that the spontaneous ERG oscillation-related component score does not correlate with either OP parameters nor DR score that reflects the risk of requiring intervention within 3 years [48].

The OPs are thought to reflect the function of the inner retina and are sensitive to changes in retinal circulation [73]. In this sense, one may wonder about the nature of spontaneous oscillations measured by non-evoked ERG. If ERG is classically conceived as the summation of local synaptic and intrinsic activities of retinal cells, the low amplitude and slow frequency oscillations recorded in the non-evoked mode may be contaminated by sources other than the retina, e.g., cardiorespiratory system and brain. In humans, with our setup setting, breathing is too slow (~0.2–0.3 Hz) [74] to be detected by ERG, as is eye movement (0.26 Hz) [75], but cardiac (0.9–1.4 Hz) [76] and brain activities may be. Also, in stably anesthetized mice, respiration is in the 0.9–1.08 Hz range, while cardiac rhythm lies between 5 and 7.5 Hz. In ketamine-xylasine-anesthetized rats, eye movement shows two frequency components, 1 and 12.2 Hz, the former correlating with respiration frequency [77]. We cannot ascertain that the full range of spontaneous oscillations measured by non-evoked ERG originates in the retina, but it is plausible that at least part of these signals come from the retina because in healthy adult retinas, several types of neurons have been reported to spontaneously oscillate, and this, within a range of frequencies from 0.7 to >10 Hz. In particular, spontaneous $Ca^{2+}$-dependent membrane oscillations have been recorded in bipolar cell axon terminals [24, 25, 78] and if the consequent pulsatile release of neurotransmitter drives rhythmic activity in post-synaptic neurons, including amacrine and ganglion cells [23, 79], different types of amacrine cells are also able to produce

intrinsic oscillatory activity [26, 27]. Low-amplitude oscillations have even been recorded in starbust amacrine cells [23]. The presence of these intrinsic oscillators in the inner retina [23–27, 78, 79], the fact that retinal neurons can be electrically coupled [80], and the large-scale network interactions happening in the retina [22] are likely to result in spontaneous fluctuations of the field potential [81] in this tissue. Consistent with the existence of bipolar/amacrine cell oscillators and our findings showing that spontaneous activity is altered in the high-fat diet-induced obesity, spontaneous obesity, and streptozotocin-induced type 1 diabetes models, inner retinal deficits have been detected at the onset of diabetes [33, 82, 83]. Overall, we found reduced peak frequencies of the low range activities (0.2–2.5 Hz), which is agreement with decreased inhibition in the early diabetic retina [33, 84–88]. Also in favor of part of the spontaneous oscillations detected by the non-evoked ERG being produced in the retina is the recent finding that DR can be detected by machine learning processing of electrooculogram (EOG) signal—corresponding to the potential difference between cornea and retina—[89].

Though studies of spontaneous retinal activity are sparse in the adult and its functional role is yet to be fully understood [90], our results support that the 0.3–40 Hz activity is relevant for the disease process. Our data show that a single model is able to distinguish disease cases from control cases under disease-relevant conditions that range from rodent models to patients. This suggests that the predictive content of spontaneous ERG signals is conserved in mammals and robust. Our PCA analysis adds to this view, because it showed that the normalized powers of the slowest oscillations (0.4–0.7 Hz) were the most important variables in separating disease cases from controls. Nevertheless, changes in spontaneous ERG oscillations of higher variability were insufficient to separate the groups of interest using PCA. Compared to the PCA that does the separation in a single step, the Random Forest model does it in several stages, that is, it first classifies the disease and control cases and then, in the disease category, classifies the four groups of interest, which surely contributes to its performance compared to that of PCA. There is also the algebraic limitation of PCA which cannot process more variables (823 in total) than the number of individuals (375, Tables 1 and 2). We do not exclude the possibility that the frequency variables analyzed by PCA may explain the multigroup separation once the number of participants exceeds the number of variables. The statistical analyses we undertook for an explanation gave us some hints: the normalized power of the slowest oscillations (from 0.3 to 0.5 Hz) and the peak frequency of the 20–40 Hz band did not correlate with the power of the 0.6–0.8 Hz components or with the AUC of the 0.3–2 and 0.3–40 Hz bands. However, we do not have yet enough data to understand in details how our predictive model uses spontaneous ERG data and the fact that it surpassed so-called "black box" algorithms such as deep learning, does not yet represent an advantage in terms of intelligibility of our model [91]. Even though we cannot explain yet the specific loadings of each variable and we do not exclude the contribution of other variables, our findings support a very tight relationship between systemic metabolic changes and retinal function [92].

Our predictive diagnostic system meets the Food and Drug Administration criteria in terms of precision ($\geq$ 85%), sensitivity ($\geq$ 85%), and specificity ($\geq$ 82.5%) for clinical validation of diagnostic tools in retina [93, 94] and the validated ROC of 0.719 alsmot indicates excellent clinical accuracy [95]. It performed as good as experts in identifying disease cases, but without the need for invasive study data. Our PCA data abound in this direction since additional variables to the spontaneous oscillations of the ERG such as VLDL, body weight, diastolic blood pressure and waist circumference, are necessary to better separate the control from disease groups. Importantly, accuracy is maintained when using signals from different sensors, which also accounts for the robustness of our system. It is well known that data from different sources penalize the performance of prediction models [96]. Moreover, the fact that spontaneous ERG protocols could be developed on three commercial sensors and that the

predictive potential of these signals is shown here illustrates that our system is flexible and can be adapted to any ERG apparatus. Also, we found that predictions are better with photopic recordings, which is advantageous in clinical practice because they take much less time than the dark-adapted ones. Based on our practical experience, we specify that 2 to 5 minutes of high signal/noise ratio recordings are enough to obtain the spontaneous 1-minute ERG sequences that, according to our data, are informative. Furthermore, the notion that the spontaneous oscillations detected by non-evoked ERG are the most precociously altered functional parameter in prediabetic stages is highly relevant in clinical practice, because OPs can only be extracted from scotopic ERG [47]. Therefore, having access to such a parameter without the need for adaptation to darkness and light flash represents a definite advantage for the patient (no pupillary dilalation, faster examination [47]). For the above and because it considers existing resources in terms of ERG device, works with completely non-invasive, portatile and cost-effective ERG devices, and does not require ophthalmological experts, our system has easy applicability in clinical settings. In our hands, the net monetary cost of a non-evoked ERG test using a portatile ERG device is less than 15 USD, and it should be noted that portable, non-mydriatic devices are about three times cheaper than desktop devices.

The clinically relevant information our system contributes to, is to detect people with preventable risk factors for type 2 diabetes. Even though the Diabetes Prevention Program recently reported that interventions that delay the development of type 2 diabetes in those at risk (overweight/obese with dysglycemia) do not reduce the subsequent prevalence of DR [97], this concerns people who have progressed to diabetes. In cases where intensive lifestyle intervention and metformin managed to prevent progression to type 2 diabetes, DR does not happen [97]. In this regard, it is important to recognize that lifestyle interventions must be carried out with considerable involvement of clinicians and that acheiving awareness among patients and family relatives is challenging.

## Limitations and future work

The advantages of our predictive diagnostic system—such as increased objectivity and efficiency in determining early risk factors for type 2 diabetes, and thereby DR, by the Random Forest system compared with health-care professionals, higher referral adherence from real-time point-of-care screening recommendations [60], more efficient resource allocation towards prevention and treatments due to the Random Forest system offloading tasks from human graders, and reduction in the prevalence of RD in the mid to long term—are implied but unproven. Also, although translation of our system into clinics seems feasible, real-world testing and acceptation is in its infancy. We have provided external validation of our system, but future data from multiethnic populations should consolidate the validation of our model. In the same line, machine learning models have the disadvantage of dealing with the problem on which they were trained. If our findings in the streptozotocin-induced diabetes model suggest that our predictive model may be useful for disease follow-up, testing more severe grades of DR and other retinal disorders, is needed to provide a multiclass prediction model with acceptable performance in an unselected population. This will also help clarify if altered spontaneous ERG oscillations are a general biomarker of neurodegenerative retinal disfunction.

An additional crucial step is to compare our system with more gold standards of early DR, like the multifocal ERG implicit time [98], and to determine whether the spontaneous 0.1–10 Hz oscillations in rodents and between 0.3–40 Hz in humans respond to a therapeutic intervention [99]. Showing altered peak frequencies and high degrees of prediction of these features only suggests that they are necessary for the early process of the diabetic eye desease [100, 101]. Deciphering pathological mechanisms responsible for these alterations will be highly

informative about early etiology of DR and may benefit the development of therapeutic options. Additional work is also needed to address the mechanisms that govern the slow spontaneous ERG signal.

## Conclusions

Our ultimate goal is to provide an effective, large-scale, easy, and affordable screening method that identifies asymptomatic patients in avoidable stages of type 2 diabetes to enable personalized preventive action. The state-of-the-art performance of our unique approach will likely contribute to improving the reputation of ERG [102], putting it at the right place in the clinical scene, as a quick, easy to administer and interpret, and relevant tool for screening not only ocular diseases, but also preventable risk factors for type 2 diabetes.

## Supporting information

**S1 Fig. Metabolic follow-up of animal models and oscillatory potential analysis.** A, Follow-up of body weight and blood glucose levels, glucose tolerance test, and insulin tolerance test in control-diet ($n = 75$) and high-fat diet-fed ($n = 75$) mice for 12 weeks. B, Body weight, blood glucose levels, glucose tolerance test, and insulin tolerance test in lean ($n = 20$) and spontaneously obese ($n = 20$) *Neotomodon alstoni* mice. In both models, mice have higher glycemia than control mice at every time point ($P < 0.05$), suggesting reduced insulin sensitivity. This was confirmed by insulin tolerance tests, which showed a lower fall in blood glucose in response to insulin in obese mice as compared with control mice. C, Blood glucose level follow-up in rats after 4, 6, 8, or 12 weeks of streptozotocin ($n = 40$) and vehicle ($n = 40$) treatment. Values, mean ± s.d. * indicates $P$ values < 0.05 determined by a two-sample Student's *t*-test in B (body weight and glycemia) and by a mixed ANOVA followed by Bonferroni test everywhere else. D, Illustrative ERG (top) and oscillatory potentials (OP, bottom) in control-diet and high-fat diet-fed mice for 12 weeks, measured in response to a light flash of 7.72 (cd. s)/m$^2$ (arrow) under dark-adapted conditions. E, Temporal monitoring (0 to 12 weeks, as indicated) of the average amplitude and implicit time of OP1, OP2, OP3, and OP4 in control ($n = 10$) and high-fat diet fed mice ($n = 12$) under dark-adapted conditions at increasing light intensities (0.02, 0.24, 2.45, and 7.72 (cd.s)/m$^2$. CD, control diet. HFD, high-fat diet. n.s., not significant ($P > 0.05$).
(TIF)

**S2 Fig. Oscillatory potentials and DR score analysis in people with early risk factors for DR, and correlation with slow frequency spontaneous ERG oscillation power.** A, Illustrative recording from the ISCEV DA 3 ERG protocol in a control case. N1-P1 and P1-N2 B, amplitude and C, peak time ratio in control (metabolically healthy, $n = 8$) and disease (OW, $n = 5$; obese, $n = 2$; MetS, $n = 6$; and diabetes with no DR, $n = 61$) groups. D, DR score in control (metabolically healthy, $n = 74$) and disease (OW, $n = 22$; obese, $n = 18$; MetS, $n = 88$; and diabetes with no DR, $n = 109$) groups. B-D, Graphs show mean ± confidence interval. Correlation analysis between the E, N1-P1 or P1-N2 amplitude ratio, F, N1-P1 or P1-N2 peak time ratio, and G, DR score with the spontaneous oscillation (SO) PCA score (detailed in Methods) in the groups of interest.
(TIF)

**S3 Fig. Random forest model performs better than support vector machine algorithms and when it primarily uses spontaneous photopic ERG of 60-s duration in humans.** A, ROC curves for both linear and radial svm algorithms. B, Performance parameters for the random forest model using power spectra from photopic or mesopic ERGs of 10, 30 or 60 s. C,

ROC curves for the random forest model using power spectra from photopic, mesopic or combined photopic and mesopic ERGs of 60 s. D, Corresponding performance parameters. All data correspond to binary classification between control and disease cases. Controls are constituted by metabolically healthy subjects ($n = 62$) and the disease group by patients with overweight ($n = 41$), obesity ($n = 16$), metabolic syndrome ($n = 55$), and diabetes with no DR ($n = 63$).
(TIF)

## Acknowledgments

We thank all volunteers under the care of the AEPC, IMO, and INDEREB. We thank Edgar Morales, Oliver Becerra, Javier Ledezma, Marco Arieli Herrera Valdez, Adriana Petriz, Susan Pérez Salazar, Jesús Chávez Baldera, and Dulce María Soria Lara for their participation in data acquisition and/or processing. We thank Edith Espino, Marina Ramírez Romero, Martín García, and Alejandra Castilla for their technical assistance, and Jessica Norris for critically editing the manuscript. We thank the many physicians, technicians, and administrators of the IMO, APEC, INDEREB, and ENES León who worked on the data collection. We thank Roger D. Traub M.D. for helpful discussions, Pavel Rueda Orozco Ph.D. and Hugo Merchant Ph.D. for insightful comments, as well as Ataúlfo Martínez Torres and Ricardo Miledí† († Deceased).

## Author Contributions

**Conceptualization:** Stéphanie C. Thébault.

**Data curation:** Ramsés Noguez Imm, Julio Muñoz-Benitez, Diego Medina, Everardo Barcenas, Guillermo Molero-Castillo, Pamela Reyes-Ortega, Jorge Armando Hughes-Cano, Leticia Medrano-Gracia, Luis Fernando Hernández-Zimbrón, Juan Fernando Rubio Mijangos, Stéphanie C. Thébault.

**Formal analysis:** Ramsés Noguez Imm, Julio Muñoz-Benitez, Diego Medina, Pamela Reyes-Ortega, Leticia Medrano-Gracia, Gerardo Rojas-Piloni, Stéphanie C. Thébault.

**Funding acquisition:** Stéphanie C. Thébault.

**Investigation:** Ramsés Noguez Imm, Julio Muñoz-Benitez, Diego Medina, Jorge Armando Hughes-Cano, Stéphanie C. Thébault.

**Methodology:** Ramsés Noguez Imm, Julio Muñoz-Benitez, Everardo Barcenas, Guillermo Molero-Castillo, Leticia Medrano-Gracia, Gerardo Rojas-Piloni, Stéphanie C. Thébault.

**Project administration:** Luis Fernando Hernández-Zimbrón, Ellery López-Star, Van Charles Lansingh, Stéphanie C. Thébault.

**Resources:** Manuel Miranda-Anaya, Gerardo Rojas-Piloni, Hugo Quiroz-Mercado, Luis Fernando Hernández-Zimbrón, Elisa Denisse Fajardo-Cruz, Ezequiel Ferreyra-Severo, Renata García-Franco, Juan Fernando Rubio Mijangos, Ellery López-Star, Marlon García-Roa, Van Charles Lansingh, Stéphanie C. Thébault.

**Software:** Julio Muñoz-Benitez, Everardo Barcenas, Guillermo Molero-Castillo.

**Supervision:** Everardo Barcenas, Luis Fernando Hernández-Zimbrón, Renata García-Franco, Marlon García-Roa, Van Charles Lansingh, Stéphanie C. Thébault.

**Validation:** Julio Muñoz-Benitez, Everardo Barcenas, Guillermo Molero-Castillo, Stéphanie C. Thébault.

**Visualization:** Ramsés Noguez Imm, Julio Muñoz-Benitez, Diego Medina, Everardo Barcenas, Stéphanie C. Thébault.

**Writing – original draft:** Stéphanie C. Thébault.

**Writing – review & editing:** Ramsés Noguez Imm, Julio Muñoz-Benitez, Diego Medina, Everardo Barcenas, Guillermo Molero-Castillo, Gerardo Rojas-Piloni, Stéphanie C. Thébault.

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
