## [Editor Report · Decision Letter 0]

30 Aug 2022

PONE-D-22-22890Preventable risk factors for type 2 diabetes can be detected using noninvasive spontaneous electroretinogram signalsPLOS ONE

Dear Dr. Thébault,

Thank you for submitting your manuscript to PLOS ONE. After careful consideration, we feel that it has merit but does not fully meet PLOS ONE’s publication criteria as it currently stands. Therefore, we invite you to submit a revised version of the manuscript that addresses the points raised during the review process.

ACADEMIC EDITOR: Please re-order the manuscript's structure into: Abstract - Introduction - Methods - Results - Discussions - Conclusion. Thank you!

We look forward to receiving your revised manuscript.

Kind regards,

Tri Juli Edi Tarigan, Ph.D

Academic Editor

PLOS ONE

Journal Requirements:

2. To comply with PLOS ONE submissions requirements, in your Methods section, please provide additional information regarding the experiments involving animals and ensure you have included details on (1) methods of sacrifice, and (2) efforts to alleviate suffering.

3. Please include your tables as part of your main manuscript and remove the individual files. Please note that supplementary tables (should remain/ be uploaded) as separate "supporting information" files.

5. We noted in your submission details that a portion of your manuscript may have been presented or published elsewhere. [https://doi.org/10.1101/2022.06.26.22276881] Please clarify whether this [conference proceeding or publication] was peer-reviewed and formally published. If this work was previously peer-reviewed and published, in the cover letter please provide the reason that this work does not constitute dual publication and should be included in the current manuscript.

---

## [Author Response · Author response to Decision Letter 0]

12 Sep 2022

All editor comments have been addressed in the rebuttal letter.

---

## [Editor Report · Decision Letter 1]

16 Nov 2022

Preventable risk factors for type 2 diabetes can be detected using noninvasive spontaneous electroretinogram signals

PONE-D-22-22890R1

Dear Dr. Thébault,

We’re pleased to inform you that your manuscript has been judged scientifically suitable for publication and will be formally accepted for publication once it meets all outstanding technical requirements.

Kind regards,

Steven Barnes

Academic Editor

PLOS ONE

Additional Editor Comments (optional):

Revisions are right on target and satisfactorily address all concerns of the previous Editorial opinion.

Sorry for the delay. Being the replacement Editor, I just took this over. I find this to be an excellent paper on an important issue.
---

## [Editor Report · Acceptance letter]

18 Nov 2022

PONE-D-22-22890R1 

Preventable risk factors for type 2 diabetes can be detected using noninvasive spontaneous electroretinogram signals 

Dear Dr. Thébault:

I'm pleased to inform you that your manuscript has been deemed suitable for publication in PLOS ONE. Congratulations! Your manuscript is now with our production department. 

Kind regards, 

on behalf of

Dr. Steven Barnes 

Academic Editor

PLOS ONE